# Mindfulness and cardiovascular health: Qualitative findings on mechanisms from the mindfulness-based blood pressure reduction (MB-BP) study

William R. Nardi[1,2]*, Abigail Harrison[2], Frances B. Saadeh[1], Julie Webb[2], Anna E. Wentz[1], Eric B. Loucks[1,2,3]

1 Department of Epidemiology, Brown University School of Public Health, Providence, Rhode Island, United States of America, 2 Department of Behavioral Sciences, Brown University School of Public Health, Providence, Rhode Island, United States of America, 3 Department of Medicine, The Warren Alpert Medical School of Brown University, Providence, Rhode Island, United States of America

* william_nardi@brown.edu

**Data Availability Statement:** The datasets presented in this article are available upon request because of sensitive and potentially identifying

## Abstract

### Background

Mindfulness-based programs hold promise for improving cardiovascular health (e.g. physical activity, diet, blood pressure). However, despite theoretical frameworks proposed, no studies have reported qualitative findings on how study participants themselves believe mindfulness-based programs improved their cardiovascular health. With an emphasis on in-depth, open-ended investigation, qualitative methods are well suited to explore the mechanisms underlying health outcomes. The objective of this qualitative study was to explore the mechanisms through which the mindfulness-based program, Mindfulness-Based Blood Pressure Reduction (MB-BP), may influence cardiovascular health.

### Methods

This qualitative study was conducted as part of a Stage 1 single arm trial with one-year follow-up. The MB-BP curriculum was adapted from Mindfulness-Based Stress Reduction to direct participants' mindfulness skills towards modifiable determinants of blood pressure. Four focus group discussions were conducted (N = 19 participants), and seven additional participants were selected for in-depth interviews. Data analysis was conducted using the standard approach of thematic analysis. Following double-coding of audio-recorded transcripts, four members of the study team engaged in an iterative process of data analysis and interpretation.

### Results

Participants identified self-awareness, attention control, and emotion regulation as key mechanisms that led to improvements in cardiovascular health. Within these broader themes, many participants detailed a process beginning with increased self-awareness to

information from this qualitative research, including original interviews, cannot be provided due to ethical restrictions. Requests to access the datasets should be directed to the Brown University Data Repository at ResearchData@Brown.edu. Study materials and analysis documentation are available through the the following link https://doi.org/10.26300/1jxn-b671.

**Funding:** EL received funding for the study. This study was supported by the National Institutes of Health (NIH) Science of Behavior Change Common Fund Program through an award administered by the National Center for Complementary and Integrative Health (UH2AT009145, UH3AT009145). The study can be found registered under ClinicalTrials.gov Identifier: NCT03124446. The funders had no role in study design, data collection or analysis, decision to publish, or preparation of the manuscript.

**Competing interests:** The authors have read the journal's policy and have the following potential competing interests: EC is the Director of the Mindfulness Center at Brown University. The Mindfulness Center, a non-profit entity, has an Education Unit that provides mindfulness-based program delivery to the general public for fees. Dr. Loucks's salary is not tied to quantity, or content, of programs offered through the Mindfulness Center, and this therefore does not alter our adherence to PLOS ONE policies on sharing data and materials. There are no patents, products in development or marketed products associated with this research to declare.

sustain attention and regulate emotions. Many also explained that the specific relationship between self-awareness and emotion regulation enabled them to respond more skillfully to stressors. In a secondary sub-theme, participants suggested that higher self-awareness helped them engage in positive health behaviors (e.g. healthier dietary choices).

## Conclusion

Qualitative analyses suggest that MB-BP mindfulness practices allowed participants to engage more effectively in self-regulation skills and behaviors lowering cardiovascular disease risk, which supports recent theory. Results are consistent with quantitative mechanistic findings showing emotion regulation, perceived stress, interoceptive awareness, and attention control are influenced by MB-BP.

## Introduction

Major determinants of cardiovascular disease (CVD) such as diet, physical activity, and hypertension, have been understood for decades [1]. Effective interventions have been developed, including behavioral, surgical, pharmacological, health systems, and policies [1–3]. CVD mortality has dropped by about 70% in recent decades because of these developments, however it remains the number one cause of death in the United States and world-wide [1, 4]. To further reduce CVD, either new classes of interventions are required, or we need to develop enhanced approaches to improve medical regimen adherence to existing evidence-based treatment and prevention strategies.

Mindfulness-based programs (MBPs) offer promise in this regard. A recently developed MBP, Mindfulness-Based Blood Pressure Reduction (MB-BP), demonstrated in Stage 1 findings that for those out of medical regimen adherence for blood pressure (BP) (SBP>140 mmHg), systolic BP was reduced by 15 mmHg by one-year follow-up [5]. These findings added to a systematic review and meta-analysis suggesting plausible effects of mindfulness-based programs on BP [6]. A systematic review and meta-analysis on mindfulness-based programs for weight loss demonstrated small to medium effect sizes in clinical trials [7]. Evidence on other CVD risk factors such as diet and physical activity is emerging [8, 9].

Multiple theories have described hypothesized mediators by which mindfulness could influence cardiovascular health and overall well-being. Major proposed mechanisms include decentering, self-compassion, acceptance, self-awareness, attention control, and emotion regulation [9–13]. Rigorous testing of the mechanisms using mediation analyses is lacking [12, 13]. The theoretical framework often applied to cardiovascular health posits three main self-regulatory pathways as important mediators between mindfulness meditation and behavioral determinants of CVD: attention control, self-awareness, and emotion regulation (**Fig 1**) [9]. *Attention control* involves maintaining present moment attention on the topic of desired focus. Improved abilities to hold attention to positive and negative sensations associated with CVD risk factors such as smoking, overeating, stress reactivity, sedentary activities, or medication adherence could be important. Holding attention to one's positive or negative relationship with these risk factors, rather than distracting away from them, can allow time and space for insights to arise that could lead to behavior change [9, 14–16]. *Self-awareness* refers to one's ability to be aware of internal thoughts, feelings, and physical sensations [9]. Greater self-awareness of thoughts, emotions, and physical sensations that are present before, during, and

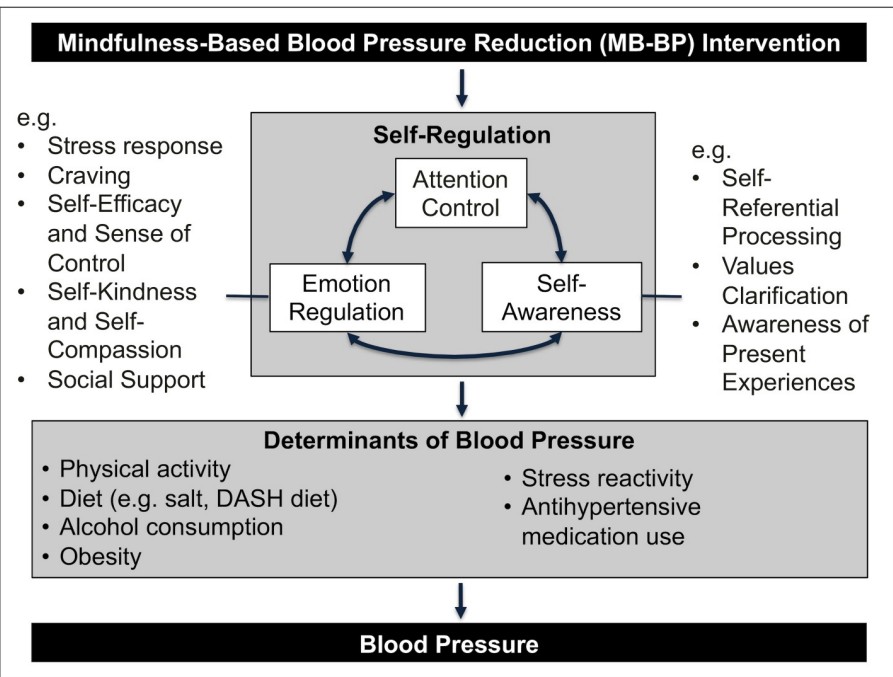

**Fig 1. Theoretical framework of mechanisms through which Mindfulness-Based Blood Pressure Reduction (MB-BP) program may influence blood pressure.**

after engagement with CVD risk factor behaviors (e.g. eating healthy foods, engaging in physical exercise, excessive alcohol consumption, cigarette smoking) may enable greater appreciation of positive experiences, and awareness of negative experiences, that occur as a result of engaging in these health behaviors. With mindfulness, individuals can be trained to become aware that the present moment is influenced by prior moments, including their diet, alcohol consumption, smoking, and physical activity [17]. Greater awareness of positive and negative experiences of these behaviors may allow for deeper insights, and improve health outcomes [9]. *Emotion regulation* is the ability to influence emotions that are experienced, including duration, intensity, and how they are acted on [18]. Improvements in emotion regulation strategies are associated with decreases in emotional interference to undesirable stimuli, reduced physiological reactivity, and more efficient return to emotional baseline after a stressful event [10]. All of these emotion regulation outcomes may foster abilities to choose healthier responses to emotionally laden situations, such as through reduced stress reactivity or stress-related behaviors such as smoking, alcohol consumption, emotional eating, and sedentary behaviors [9].

MB-BP is customized to people with elevated BP. It was designed to improve the self-regulatory mechanisms above (**Fig 1**), and then direct those skills towards participants' relationships with evidence-based determinants of blood pressure, including diet, physical activity, alcohol, stress reactivity and antihypertensive medication adherence [5]. Quantitative findings supported these mechanisms showing significant improvements in attention control (Sustained Attention to Response Task correct no-go score, p<0.001), self-awareness (Multidimensional Assessment of Interceptive Awareness score, p<0.001), and emotion regulation (Difficulties in Emotion Regulation score, p = 0.02; Perceived Stress Scale score, p = 0.01) at one year follow-up [5].

When evaluating mechanisms of mindfulness interventions, scientists often create *a priori* hypothesized mechanisms that are then tested quantitatively. However, in many ways, as

individuals who completed the program, the participants themselves are the experts about how the MBP influenced their well-being. Despite theoretical frameworks proposed, and preliminary supportive quantitative mechanistic findings, no studies to our knowledge have reported qualitative findings on how study participants perceive and feel that MBPs improved their cardiovascular health. The explanatory nature of qualitative methods allows them to contribute to in-depth investigations of mechanisms beyond *a priori* hypotheses, as well as generate new theories and hypotheses. In fact, qualitative methods enhance the data being collected by enabling investigators to compare their perceptions with the perceptions of participants [19].

Consequently, the objective of this qualitative study was to investigate the mechanisms by which the MB-BP program may influence cardiovascular health, utilizing focus group discussions and in-depth interviews. Specifically, this study presents findings from focus group discussions and in-depth interviews on three topics: (1) The utility of mindfulness practices for BP reduction; (2) Consistency of reported findings with the proposed theoretical framework; and (3) Emergent themes investigating novel mechanisms that contributed to behavior change.

## Methods

The study protocol was approved by the institutional review board at Brown University (protocol #1412001171) on September 3, 2015. Participants provided written informed consent.

### Stage 1 study sample description

Participants in the main trial were recruited and assessed during 2016–2017. Most participants were recruited via flyers and recruitment cards distributed throughout Rhode Island and Massachusetts (43% of participants), family/friend referrals (19%), former/current participant referrals (16%), and referrals from primary care practitioners or other health care professionals (16%).

Inclusion criteria were: (1) Hypertension/prehypertension (SBP≥120 mmHg systolic or DBP ≥80 mmHg or prescribed antihypertensive medication for treatment of hypertension); (2) Able to speak, read, and write in English; (3) All adults (≥18 years of age), genders and racial/ethnic groups were eligible to be included.

Exclusion criteria were: (1) Current regular meditation practice (>once/week); (2) serious medical illness precluding regular class attendance; (3) current substance abuse, suicidal ideation, or eating disorder; or (4) history of bipolar or psychotic disorders or self-injurious behaviors. These participants were excluded following standard guidelines because of risk for disrupting group participation, requiring additional or specialized treatment beyond capacity of this study, or already participating in practices similar to the intervention [20]. The study protocol was approved by the institutional review board at Brown University (protocol #1412001171 on September 3, 2015. Participants provided informed consent. Data were collected as part of the MB-BP Study (Clinicaltrials.gov #NCT02702258), described in more detail elsewhere [5].

### The qualitative study

Purposive sampling of intervention participants was used to recruit for the qualitative study [21]. Intervention study participants were contacted by members of the research team to assess their willingness to participate. Participation was voluntary and independent of participation in the intervention trial. All study participants who completed the MB-BP program (n = 48) were eligible to participate.

First, three focus group discussions (FGDs) were conducted including 19 participants (FGD 1, n = 4; FGD 2, n = 8; FGD 3, n = 7). The FGDs were conducted by a co-author (AH), an experienced qualitative researcher, assisted by a note taker (AW). Neither had a relationship with protocol development, implementation, intervention assessments, or contact with participants prior to the FGD. The FGDs were audio recorded and held in a private room at the Brown University School of Public Health in Providence, RI. FGDs occurred within two months after completion of the intervention. Each session lasted between 1.5–2 hours and was conducted in accordance with an approved focus group protocol see **S1 File**.

FGD participants were provided with food and refreshments but were not compensated for participation.

Seven semi-structured, in-depth interviews (IDIs) were conducted with participants who declined participating in the focus group discussions, in order to foster a more representative sample, and to more effectively capture individual perspectives and experiences with the intervention [19]. They were selected using a randomized list of possible candidates by a senior member of the research staff. Those selected were then followed up to inquire about whether they would be willing to participate in an in-depth interview. Ten participants were contacted with three refusals in order to recruit the seven individuals included in the IDIs. All of the IDIs were conducted by phone, by a senior member of the research team (WN). Participants were reimbursed with a gift card ($25) for participating in the 25-minute IDI.

## In-depth interviews (IDI)

The IDIs were conducted by a member of the study staff trained in qualitative methods under the guidance of the consulting qualitative expert (AH). The researcher conducting the interviews was familiar with MBPs and interested in conducting and evaluating MBPs. The interviews were conducted using Zoom (Version # 4.1.24407.0507, San Jose, CA.) software and audio recorded. The interviews were conducted in accordance with an approved protocol see **S2 File**.

## Intervention description

This study adapted Mindfulness-Based Stress Reduction (MBSR) for participants with prehypertension/hypertension, creating MB-BP. Specifically, MB-BP is based on, and time-matched to, the standardized MBSR intervention which consists of eight 2.5-hour weekly group sessions, and a 7.5-hour one-day sessions, and is described elsewhere [5, 22–25]. MB-BP and MBSR contain similar instruction and practices in mindfulness meditation and conversations about stress and coping.

The unique areas of MB-BP are education on hypertension risk factors, hypertension health effects, and specific mindfulness modules focused on awareness of diet, physical activity, medication adherence, alcohol consumption, stress, and social support for behavior change. MB-BP builds a foundation of mindfulness skills (e.g. meditation, yoga, self-awareness, attention control, emotion regulation; **Fig 1**) through the modified MBSR curriculum. MB-BP then directs those skills towards participants' relationship with their risk factors for hypertension. Details concerning study design, intervention assessments, and preliminary acceptability and feasibility results as well as customizations of MB-BP from MBSR are found in further detail elsewhere [17]. The curriculum guide and MB-BP instructor certification program can be accessed by contacting the principal investigator (EL).

## Data analytical approach

Trained research assistants reviewed audio recordings and transcribed recordings verbatim. The data was collected as part of a larger mixed methods study and was analyzed using

thematic analysis [5, 26–28]. Thematic analysis was considered the most appropriate analytical approach for the semi-structured data from the focus group discussion and in-depth interviews.

Both FGD and IDI transcripts were double-coded by two members of the research team who then identified both *a priori* and emergent codes by two members of the research team (WN, JW) using NVivo v.11 [27, 28]. Cross checks for coding consistency were performed by a third member of the research team (AW). Codes with definitions are listed in the qualitative codebook (see **S1 Table**). The data are available from the corresponding author upon request with ethics approval.

The team met under the supervision of the qualitative expert (AH) during a series of three in-person meetings to: (1) cluster FGD coding into broad themes, (2) discuss themes and additional qualitative work needed, and (3) refine FGD and IDI themes into main themes and sub-themes. During the iterative process of data analysis for the FGDs, the team found that saturation had not been achieved. In-depth interviews were planned to gain further insights into the main research questions, as well as to gain an individual perspective to enhance the group-based collective understanding of the FGDs. Videoconference- and phone-based IDI's provide an additional benefit to reduce possible biases associated with focus group participation, where it is plausible that those who had less positive experiences in the MB-BP course may have declined coming to the in-person focus groups [29]. In addition to the importance of the individual perspective gained from the IDIs, participant burden for phone/videoconference IDI's is less and participation rates are higher, and so it was thought that this could provide a more representative sample overall [29, 30]. The coding structure from the FGD was used to inform the coding of the IDI's. Discrepancies were resolved through group consensus and communications between the coders (WN, AW, JW) and the qualitative expert (AH) with input from the content expert when applicable (EL). The results of this process and the integrated thematic analysis of qualitative data from both FGDs and IDIs are presented below.

## Results

The qualitative sample (n = 26) represented over half of MB-BP Stage 1 participants (N = 48), with no significant differences in demographics, hypertension status, or determinants of hypertension, between the full MB-BP sample and the qualitative samples (**Table 1**.) The number of coding references for each topic are shown in **S1 Table**. During data analysis, three major themes emerged, which are discussed in detail below. An overview of main themes and subthemes is presented in **Table 2** with representative quotations.

### Theme one: Breath awareness, followed by body awareness, were the mindfulness practices that participants used most in their daily lives

Participants reported using a variety of mindfulness practices. However, they used breath awareness most often. Some became aware of difficult physical sensations, and used the breath practice to reduce discomfort: *"Being aware of how different things affect your body. I would see my hands and feel the tension and start to breath and calm myself down."(F2)* Another participant said that she believed that breath awareness could be used for specific physiologic change (e.g. lowered heart rate): *"The awareness of breathing and being able to kind of lower my heart rate and relax. If I sit there and focus on it, it's been helpful."(IDI4)*

Breath awareness was also used regularly during the S-T-O-P practice. In this practice participants **S**top a moment, **T**ake a breath, **O**bserve opening towards thoughts, emotions, physical sensations and then **P**roceed with an activity that will support an effective response to the experience. For one participant, *"being able to control my breathing has been very helpful—like*

**Table 1. Characteristics of study sample at baseline, stratified by study sample group.**

| Variable | MB-BP Study (n = 48) Point Estimate | Qualitative Sub-Study (Includes FGD and IDI Samples; n = 26) Point Estimate | $p^\S$ | FGD-Only Sample (n = 19) Point Estimate | IDI-Only Sample (n = 7) Point Estimate |
|---|---|---|---|---|---|
| Age, y | 60.0 | 61.5 | 0.85 | 62.4 | 59.1 |
| Race, % White | 95.8 | 92.3 | 0.51 | 94.7 | 85.7 |
| Gender, % women | 60.4 | 61.5 | 1 | 63.2 | 57.1 |
| Education, % college education | 91.7 | 92.3 | 1 | 89.5 | 100 |
| Hypertension Status | | | | | |
| Uncontrolled stage 2 hypertension,* % | 47.9 | 46.2 | 0.83 | 36.8 | 71.4 |
| Stage 1 hypertension, % | 33.3 | 34.6 | | 36.8 | 28.6 |
| Elevated blood pressure, % | 18.8 | 19.2 | | 26.3 | 0.0 |
| Taking antihypertensive medication, % | 60.4 | 53.9 | 0.93 | 52.6 | 57.1 |
| Body Mass Index, kg/m$^2$ | 28.5 | 28.0 | 0.78 | 28.6 | 26.5 |
| DASH Diet Score | 3.27 | 3.38 | 0.93 | 3.39 | 3.36 |
| Aerobic Physical Activity, % adhering to AHA physical activity guidelines† | 69.0 | 62.0 | 0.81 | 58.0 | 71.0 |
| Daily Alcohol Consumption, mean number of drinks per day | 0.65 | 0.32 | 0.11 | 0.25 | 0.51 |
| Perceived Stress Scale score | 22.7 | 22.8 | 0.99 | 22.5 | 23.4 |
| Antihypertensive medication use,% ‡ | 60.4 | 56.0 | 0.96 | 55.6 | 57.1 |

Stage 2 hypertension: SBP≥140 or DBP≥90 mmHg; Stage 1 hypertension: 130≤SBP<140 or 80≤DBP<90 mmHg; Elevated blood pressure: 120≤SBP<130 mmHg and DBP<80 mmHg.

MB-BP, Mindfulness-Based Blood Pressure Reduction; DASH, Dietary Approaches to Stop Hypertension; DBP, diastolic blood pressure; SBP, systolic blood pressure; FDG, focus group discussion; IDI, in-depth interviews.

* "uncontrolled" stage 2 hypertension reflects participants with SBP≥140 or DBP≥90 mmHg at baseline. Participants stage 1 hypertension or elevated blood pressure categories may have been in stage 2 hypertension in the past, but are currently controlled in elevated, or stage 1, hypertension due to antihypertensive medication or nonpharmacologic interventions.

† AHA guidelines, consistent with Rapid Assessment of Physical Activity, are engaging in 30 minutes or more per day of moderate physical activities 5 or more days per week, or 20 minutes or more per day of vigorous physical activities 3 or more days per week

‡ Percentage using any antihypertensive medication use at baseline determined by prescription data. N = 1 observation was missing this data.

§ P values compare the qualitative sub-sample with the trial sample not participating in either IDIs or FGDs and were calculated using ANOVA for continuous variables and the Fisher exact test for categorical variables.

*I think about, wait a minute, stop, there's actually an exercise S-T-O-P. So I stop, take a deep breath, and regulate my breathing–I find it so helpful in life."(F1)* Another participant detailed how this practice helped her navigate complicated social relationships. The participant was able to respond to stressful events more effectively: *"it's changed how I interact with her in particular, um. I think that it's huge you know–the fact that it could have that kind of effect on a relationship that, that I very much treasure [. . .] it helps me stop and think about what I'm doing, and it's been a positive experience for both of us."(IDI3)*

However, another participant indicated that the breath awareness alone was enough, and although he appreciated that S-T-O-P exercise, he used it less than the breath awareness: *"I found those [breath awareness meditations] probably more helpful than the S-T-O-P."(IDI2)* Overall, breath awareness was the most frequently used mindfulness practice, and often within the S-T-O-P tool.

Body scanning was also widely used. One participant explained that she became aware of tension or physical sensations in specific areas of her body and this practice allowed her to notice and resolve tension in the moment: *"realizing through the day when my jaw is clenched,*

**Table 2. Qualitative themes outlining mechanisms by which MB-BP may influence blood pressure.**

| Breath & body awareness were the most applied mindfulness practices. | Representative Quotes |
|---|---|
| Breath awareness | *"I think just the awareness of breathing and being able to kind of lower my heart rate and relax. If I sit there and focus on it, it's been helpful."* (female, IDI) |
| Body awareness | *"For me the body scan. Particularly at night when you're trying to go to sleep you realize your shoulders are up, your back is tight, your jaw is tight. That's very helpful to me and realizing through the day when my jaw is clenched and just can start the breathing and do the body scan."* (female, FGD) |
| **The increase in self-awareness allowed participants to emotionally regulate.** | |
| Shift to self-kindness | *"You're aware of that stress level, and you're able to say, 'Yeah that's stress, and let's step away and back from that stress to take care of yourself, and relax."* (female, FGD) |
| Attentional redeployment to decrease reactivity | *"Taking the necessary steps to regulate it [anxiety]. Whether it be breathing or just noticing it in my body and just most of the time its stopping and breathing I can just feel the tension dissolve. I think for me that has to be helping my blood pressure."* (male, FGD) |
| **Emotion regulation strategies impact specific hypertension risk factors.** | |
| Improved stress management | *"You're aware of that stress level and your able to say 'yea that's stress' and let's step away and back away from that stress to take care of yourself and relax."* (female, FGD) |
| Improved dietary behaviors | *"Being conscious of what I eat and making responsible choices and all. I'm still working on that one and sometimes I could through periods there were several days of being mindful of that."* (male, IDI) |

IDI: In-depth Interview, FGD: Focus Group Discussion.

and just can start the breathing and do the body scanning, and as I said at our group the other day, I can mentally shake my body out."*(F3)* She also reported applying the body scan practices before bed to alleviate excessive tension. *"Particularly at night when you're trying to go to sleep, you realize your shoulders are up, your back is tight, your jaw is tight."* One participant stated that body scanning was particularly important, and she that it was a practice she relied *"on all the time now. . . it worked."(F13)*

Another agreed, stating that becoming aware of the body was a critical practice they continue to use in their daily life: *"the body scan, which I had not been exposed to before, is something that was the most. . . [pause]. . . it was the first thing that we did and it was the first thing that I remembered and which had a lasting effect and which I rely on all the time now."* (F10)

## Theme two: Participants used these practices as a way of grounding in the moment, allowing them to emotionally regulate more effectively

The increase in self-awareness of their present moment allowed participants to emotionally regulate more effectively. Practices, like breath and body awareness, increased present moment awareness allowing participants to become more conscious of physiologic sensations (i.e. interoception). By recognizing the physical sensations and associated thoughts, they could stop and choose their response to a given set of circumstances. For example, a participant indicated that she used breath awareness when she was upset to reduce stress and her BP: *"I feel like a combination of being more aware, you know you're in the car sometimes and something,*

*you know, gets you upset. Just being able to take a few deep breaths, which I think affect definitely, you know, helps to lower the blood pressure."(F9)*

She stated breath awareness provided space for her in present moment experience, allowing her to choose her response instead of reacting: *"[I] would never pay attention before, now you're in the moment. And you have control to a degree, you can get yourself together better, to react in a better way [. . .] just take a deep breath, you know, a couple of deep breaths, and think about what's going on right now and how can you react to it."(F9)*

Another outlined how awareness was a tool that allowed him to become aware of the stressful situation and choose how to work with resultant emotions, specifically anger: *"the phrase was to be aware, I'm aware of my anger and all of that sort of relates into this tool of awareness that we have been given and all this sort of relates to the breath. So that you're aware of that stress level and your able to say yea that's stress and let's step away and back from that stress to take care of yourself and relax." (F4)*

**Theme 2a: Participants used self-kindness practices for emotion regulation.** With this increased ability to choose, many participants chose to apply self-kindness rather than reacting with self-judgement in their daily lives. Participants explained that mindfulness allowed them more time before responding to a stressor to act from a place of kindness for themselves *"I have alarm bells go off in my head, and I know, I need to [. . .] stop, take a breath, opening towards me, and proceed. And, all of his reminders to us to be. . . kind. . . it was being kind to yourself."(F6)* One participant described being able to notice physical signs of stress and take care of herself: *"you're aware of that stress level, and you're able to say, 'Yeah that's stress, and let's step away and back from that stress to take care of yourself, and relax.'"(F4)* Another participant explained self-kindness took the form of increased gratitude *"something is going wrong in my life and I just stop and say I've got shoes, I've got my glasses, I've got my car, and I have all these things. And when you stop and think about it, the idea that you've got to have this and you've got to have this to be happy–it's not true at all."(F3)*

There was an also an indication that self-kindness impacted interpersonal relationships. A participant reported increased ability to stay present with another person, listening from a place of kindness: *"Listening to what they were saying and also at the same time getting a whole emotional response from the other person and from within ourselves. So I can. . . for me, that was a beautiful learning–a whole other level of perceiving emotions."(F8)* Another participant, who was providing care to her husband, developed kindness for herself and others: *"I was just beyond stressed, and just backing off and taking care of myself, and going to the class. I have loving-kindness now, and I can feel it every day. And then today I thought, "Oh, that's loving-kindness," and I thought, "but I don't practice it with that person." [group laughs]. But then I thought, "Oh, you know I can do it with my husband now, but I have to look at other people." Those insights are really, really valuable."(F11)*

**Theme 2b: Attentional redeployment emerged as an important emotion regulation strategy.** Participants also used attentional redeployment as an emotion regulation strategy to influence their body's reactivity, including their BP. They reported noticing stressful events and choosing to move attention towards mindfulness practices instead of ruminating on difficulties. One participant noticed anxiety and re-engaged with breath awareness, decentering from the experience: *"I get anxious easily just slowing down and realizing that"* from this point *"taking the necessary steps to regulate it, whether it be breathing, or just noticing it in my body, and just most of the time it's stopping and breathing. I can just feel the tension dissolve."(F1)* He believed that reorienting attention away from negative thoughts and sensations towards the body/breath affected his BP *"[regulating] has to be helping my BP."(F1)* Another applied re-orienting attention to improve his BP control *"it basically took that game [BP control] to a better level; a higher level."(IDI7).* He was *"aware of when my body is wound up"* and from there, he

was able to choose *"breathing the focused relaxation to let a lot of that go. To let a lot of attention, go."(IDI7)* One participant summarized that during her meditation practice, she noticed her heart racing and redirected attention as an immediate, effective response, which she now chooses to use regularly *"I could feel my heart beating, not overly fast, but it's really going. But then after a couple of minutes, I can feel everything leveling off. So I have carried that through my daily life with just stopping and breathing and thinking about how that breathing is affecting my body, because I can physically feel a difference."(F3)*

### Theme three: Participants applied emotion regulation strategies to specific hypertension risk factors: Stress and dietary choices

Most often participants used emotion regulation strategies to respond more effectively to stress. The practices (e.g. self-kindness, breath awareness, body awareness) allowed them to feel *"a lot sooner when you'd get stressed,"(F9)* and choose an accepting response to the stressor rather *"than get up and get all stressed out again. And then just think 'Uh oh, I guess that's just how life is,' and now I realize, 'No.'"(F9)* Participants agreed that they had increased their abilities to choose a response to stress. For example, *"it helped me just to be better and be more in the moment, and to stop."* and *"It was helpful in terms of me becoming more self-aware, and especially breaking patterns of responses."(F10)* One participant indicated he often chose breath exercises, stating he is *"more relaxed and aware of my doing breathing exercises. Also, I find that to be very helpful in reducing my stress level."(IDI6)* Another participant applied these exercises specifically to stress during traffic *"while I'm driving to work, instead of that being like I have to hurry up and get there, like to try to do breathing exercises in the car and relax and not worry about the traffic."(IDI4)*

Participants also modified dietary choices as a result of increased emotion regulation. One participant was able to mitigate stress and make heathier dietary choices, which became easier to maintain for longer periods of time: *"being conscious of what I eat and making responsible choices, and I'm still working on that one. And sometimes I could go through periods of several days or weeks being mindful of that."(IDI7)* This participant had been avoiding salt, but the practices helped to continue reducing salt intake: *"I find it necessary to be making, healthful choices in what I'm eating."(IDI7)* Another saw change in his food choices, as well as other risk factors for hypertension, and stated, *"[I] can feel when I'm elevated and feel like I'm super tense, and I can also really physically choose how to lower that–whether it's running, you know, doing some physical exercise, breathing, meditating, yoga, you know, or what I eat."(F5)* However, he emphasized that the increased choices had most importantly changed how he looked at food: *"the way I look at food in a lot of ways, and you know. . . There's that hesitation, that moment like, 'Oh, wait, I have a choice here, you know, maybe I won't do that.' And it's given me more ability to do that."(F5)*

There was an indication by one participant of a higher level of awareness and self-kindness while eating healthy, versus unhealthy, foods: *"reinforce thinking about how your body feels when you eat the foods that maybe aren't making your body feel good, versus the foods that are more helpful." (F6)* Healthy eating affected one participant's interpersonal relationships, allowing him to *"notice on vacation when I had too much caffeine, and when I overate, that it influenced the way that I treated other people."(F16)*

In contrast another participant explained that increasing self-awareness may have led to higher stress. This participant *"had a medical problem that surfaced just before it started that was bothering me a lot with my eye,"* and indicated that *"the breathing helped me calm down"* in response to the stress, *"but I don't think it was enough to affect my blood pressure."(F6)* In fact, he noted that he was *"a little scared though [. . .] I really never thought I had a problem with my blood pressure."* However, as a result of meditation, he *"[paid] more attention to [BP], which I*

*think I just ignored before. So it was a little upsetting to me."(F6)* This participant found mindfulness practices useful (e.g. breath awareness previous statement) but the program increased awareness of having "*a problem [hypertension]*" which was difficult to come to terms with: *"I initially said I wanted to do it [the MB-BP program] to learn about mindfulness, but then all of the sudden it felt like I was acknowledging I had a problem about something, and it was upsetting to me to have a problem. So I think it's going to take time to improve my blood pressure."(F6)*

Overall Summary. For many participants the program led to important changes in their health behaviors after becoming more self-aware and for some these changes were made in multiple areas of their lives. This is best captured by one participant who described how she was able to exert greater control over her life, stress responses, eating behaviors, and anxiety to lower her BP: "Being mindful of what makes us anxious and trying to control our reaction to what makes us anxious and, in addition, we were also taught to be mindful of what makes us happy. So, to reinforce different things that make you happy, and also with the eating, reinforce thinking about how your body feels when you eat the foods that maybe aren't making your body feel good versus the foods that are more helpful. So, I think it affected everything in my whole life."(F2).

In summary, participants indicated that the program impacted their cardiovascular health directly through becoming more aware of their present moment experiences, allowing them to make positive choices for their health behaviors, specifically minimizing stress reactivity and making healthier choices concerning their diet.

## Discussion

In this study, participants identified attention control, self-awareness, and emotion regulation as mechanisms by which MB-BP influenced their cardiovascular health. Specifically: (1) Participants most frequently used the breath and body awareness mindfulness practices; (2) Participants detailed a temporal ordering of the mechanisms, where the breath and body awareness provided a foundation by which they could then choose to more effectively regulate their emotions and (3) improved emotion regulation supported engagement in behaviors consistent with medical regimen adherence (e.g. stress response strategies, healthy dietary patterns).

The findings add richness and build on the proposed theoretical framework [9] as well as quantitative findings from the Stage 1 trial [5]. According to the theory proposed by Tang et al, [10] mindfulness meditation fosters higher levels of self-regulation through emotion regulation, self-awareness, and attention control [10, 31]. Participants' experiences in Theme 2 detail that in practical application there may be a stepwise process by which an individual first recognizes their present moment experience, is able to pause grounded in the experience of the present moment, and then choose a more effective response to their current situation. In addition, the application of self-kindness as an emotional regulation strategy is consistent with the theoretical framework while attentional redeployment is a theme that emerged from the data [9]. The evidence presented suggests that MB-BP does increase self-regulation of health behaviors, specifically related to determinants of BP (e.g. diet, physical activity, stress reactivity; Theme 3; **Fig 1**) [31]. Given the behavior change reported in the MB-BP study, both through participants' own narratives as well as the study's quantitative findings [5], these results are encouraging for hypertension risk reduction strategies.

Our findings on the importance of self-awareness are consistent with prior qualitative work [32, 33]. Self-awareness has been investigated as a critical component of other therapeutic approaches, including standard MBPs (e.g. MBSR), complementary and alternative therapies (e.g. Qigong), and traditional therapeutic approaches (e.g. physical therapy, body-oriented psychotherapy) [34]. A study analyzing diary entries of eight participants in a MBSR program

identified core themes of increased awareness, less judgment, and improved ability to re-perceive stressors in ways that improved stress reactivity [35]. More recently, eight expert teachers in mind-body therapies (e.g. Tai Chi, body-oriented psychotherapy, mindfulness-based therapies/meditation) and eight clients of these experts were invited to participate in focus groups discussing the importance of self-awareness–specifically body awareness [33]. The experts reported that self-awareness was critical to their practice, and that a goal of their therapies was to integrate self-awareness of both mind and body [33]. Their clients reported improvements in present moment awareness, awareness of negative emotions, ability to choose responses, and their relationships between self-regulation and self-care. These previously published findings are consistent with the results presented in the MB-BP study highlighting the importance of targeting self-awareness as a mechanism through which mindfulness interventions impact health behavior change [33].

The participant who related how increased self-awareness of their blood pressure made them *"a little scared"* and that the course made them aware of problem with their blood pressure that they had not accepted until this point is important to note. The participant's shift from being unaware to accepting and acknowledging the health concern is not surprising as MBPs are experiential and involve a process of inquiry where participants "learn to identify their thoughts, emotions, and sensations, recognize habitual patterns of reacting to them, and respond with greater awareness and flexibility" and this can produce unwanted effects (UE) [36]. However, the participants willingness to continue to engage in practice despite a difficult realization is consistent with previous research in that under 10% of participants elect to stop practicing after experiencing an UE and express that, despite difficulties, there are clear benefits to practice [36–38]. The preliminary evidence for MBP's regarding adverse effects is consistent with broader psychological research however critical questions regarding harms remain largely unanswered including the frequency and severity of events, the impact of intervention dose and home practice amount on harms as well as whether some participants are more or less likely to experience difficulty [36, 39, 40].

As such, moving forward it is critical to continue to identify what mechanisms underlie mindfulness-based programs for specific clinical populations and provide clarity on what clinicians should consider when recommending an MBPs. Using the current study as an example, investigating the mechanisms through which mindfulness may influence hypertension risk factors is necessary to identify their etiology, customizing these interventions to target key mechanisms of change, and removing inactive program elements that contribute to cost and time inefficiency. However, in identifying mechanisms, it is critical to determine if there is a main effect of MBPs on BP. The current stage 1 trial provides preliminary evidence investigating not just BP change (described elsewhere) [5], but also by identifying distal and proximal mechanisms through which mindfulness can impact hypertension. Importantly, qualitative MB-BP results provide clearer evidence regarding quantitative findings (e.g. associations with Dietary Approaches to Stop Hypertension diet, physical activity, alcohol consumption, stress, attention control, interoceptive awareness, emotion regulation) which were significantly improved through to one-year follow-up [5]. The qualitative findings presented suggest mediating pathways, offer new questions regarding the relationship between emotional regulation and self-awareness, and represent an important step in understanding the most effective measures for more rigorous interventions to target when investigating mindfulness and hypertension risk reduction. The identification of these mechanistic pathways provides clear next steps for researchers. Specifically, future research should investigate the most important mechanistic pathways for hypertension risk reduction and health behavior change, including Stage 2 randomized controlled trials (currently underway see Clinicaltrials.gov NCT03256890) as well comprehensive mediation analyses.

Strengths of this study include the use of a multi-method qualitative approach, using both FGDs and IDIs, which offer the reflections of participants from both group and individual perspectives. In addition, the in-depth interviews allowed for inclusion of participants who may have been hesitant to participate in FGD (e.g. because of negative views regarding the program, or time burden) to be included, thereby reducing participation bias and raising the sample size for the qualitative sub-study (i.e. 54% of the Stage 1 participants, n = 26). Furthermore, neither the research assistants conducting the FGD (XX, XX) nor the qualitative expert overseeing the coding and thematic analyses (XX) were affiliated with the mindfulness research program, thereby minimizing potential expectancy biases. Limitations included a relatively homogenous population by race/ethnicity, socioeconomic status, and geographical region, thereby limiting generalizability, although participants were fairly diverse in gender and age. Finally, the MB-BP curriculum emphasizes the theoretical framework that places attention control, emotion regulation, and self-awareness as mediators for how the program could influence participants' BP [5]. It may be that other MBPs, which are less explicit about this theoretical framework, would have different findings. However, the similar mechanistic findings in other qualitative MBP studies described above suggest possibility for consistency in mechanisms [33–35]. Replication of the findings are needed by independent research groups for this and other MBPs, as well as more diverse samples.

In conclusion, this study suggests that self-awareness, attention control, and emotion regulation are mechanisms by which MBPs improve cardiovascular health. These qualitative findings provide deeper insight and an understanding of participants' own perspectives on which aspects of the MB-BP intervention help them, and how that relates to engaging in behaviors aimed at reducing CVD risk. Understanding mechanisms is useful not only in identifying the etiology of interventions, but also adapting them to more efficiently engage with the key mechanisms of change [41]. We are at a time in history when CVD is the number-one cause of death globally, and 1.13 billion people have stage 2 hypertension, of which only one in five people have it controlled [3, 42]. It is imperative to continue exploring and developing mechanism-driven interventions building on the existing evidence base, in order to make further progress on driving down global CVD mortality.

## Supporting information

**S1 Table. Mindfulness-Based Blood Pressure Reduction study: Stage 1 qualitative codebook: File contains the qualitative codebook with operationalized coding definitions and coding frequencies both total number for each code as well as number of data sources containing the referenced code.**
(DOCX)

**S1 File. Focus group protocol.**
(DOCX)

**S2 File. In-depth interview protocol.**
(DOCX)

## Author Contributions

**Conceptualization:** William R. Nardi, Abigail Harrison, Eric B. Loucks.

**Data curation:** William R. Nardi, Abigail Harrison, Frances B. Saadeh, Julie Webb, Eric B. Loucks.

**Formal analysis:** William R. Nardi, Abigail Harrison, Julie Webb, Anna E. Wentz.

**Funding acquisition:** Eric B. Loucks.

**Investigation:** William R. Nardi, Eric B. Loucks.

**Methodology:** William R. Nardi, Abigail Harrison, Frances B. Saadeh, Anna E. Wentz.

**Project administration:** William R. Nardi, Frances B. Saadeh, Eric B. Loucks.

**Resources:** William R. Nardi, Frances B. Saadeh, Eric B. Loucks.

**Software:** William R. Nardi.

**Supervision:** Abigail Harrison, Frances B. Saadeh, Eric B. Loucks.

**Writing – original draft:** William R. Nardi, Abigail Harrison, Frances B. Saadeh, Julie Webb, Anna E. Wentz, Eric B. Loucks.

**Writing – review & editing:** William R. Nardi, Abigail Harrison, Anna E. Wentz.

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
