## [Decision Letter · Decision Letter 0]

29 Jul 2020

PONE-D-20-16708

Mindfulness and Cardiovascular Health: Qualitative Findings on Mechanisms from the Mindfulness-Based Blood Pressure Reduction (MB-BP) Study

PLOS ONE

Dear Dr. Nardi,

Thank you for submitting your manuscript to PLOS ONE. After careful consideration, we feel that it has merit but does not fully meet PLOS ONE’s publication criteria as it currently stands. Therefore, we invite you to submit a revised version of the manuscript that addresses the points raised during the review process.

This is a methodologically strong and well-written manuscript reporting interesting new qualitative findings on the possible processes underlying positive effects of mindfulness on cardiovascular health. Minor amendments to the manuscript are needed before it can be accepted for publication in PLOS ONE. I agree with the comments from Reviewer 1, please make sure to address these in your revision. I have a couple of additional comments that need addressing in the revision:

- 'craving' is included as a possible mechanisms of mindfulness on line 79, this should be corrected or clarified

- I don't think 'self-awareness' and 'interoception' are equivalent terms, 'interoception' describes sensation-related awareness whereas 'self-awareness' is a broader term involving awareness of self-construal related processes (lines 89-90), please clarify why you think the two terms can be used interchangeably or don't use them in this way

- please explain briefly why the thematic analysis approach has been chosen and support the explanation by relevant references

We look forward to receiving your revised manuscript.

Kind regards,

Dusana Dorjee

Academic Editor

PLOS ONE

Journal Requirements:

"The authors have declared that no competing interests exist. Dr. Loucks is Director of the Mindfulness Center at Brown University. The Mindfulness Center, a non-profit entity, has an Education Unit that provides mindfulness-based program delivery to the general public for fees. Dr. Loucks’s salary is not tied to quantity, or content, of programs offered through the Mindfulness Center."

Reviewers' comments:

Reviewer's Responses to Questions

**Comments to the Author**

1. Is the manuscript technically sound, and do the data support the conclusions?

Reviewer #1: Yes

2. Has the statistical analysis been performed appropriately and rigorously? 

Reviewer #1: Yes

3. Have the authors made all data underlying the findings in their manuscript fully available?

Reviewer #1: Yes

4. Is the manuscript presented in an intelligible fashion and written in standard English?

Reviewer #1: Yes

5. Review Comments to the Author

Reviewer #1: I enjoyed reading this manuscript that employed qualitative methods to gain in-depth understanding of how a mindfulness-based intervention could improve health behaviors related to elevated blood pressure. The manuscript is a joy to read, with clear and well-written text sections in a logical, easy-to-read flow. The context of the study in the phase-I trial is well explained. Overall, the study offers important information for understanding mechanisms by which the mindfulness intervention improves health behaviors that could lead to the substantial reduced BP observed in the phase I trial. The addition of the in-depth interviews to the focus groups is lovely (to reach further!). I have a few minor comments for improvement as follow:

1) Though the explanations are clear, more references for the qualitative methods used are needed. I note that 'The Qualitative Study' (lines 157-192) and the 'Data Analytic Approach' (lines 210-234) have limited or no references. Adding these references in would enhance the manuscript and provide readers with additional resources to understand the methods used (i.e., the qualitative research methodology).

2) The results are lovely, but would benefit from an additional subtitle to separate the overall summary at the end. As it is, the 3rd theme just bleeds into the overall - an 'Overall findings' or 'Summary of Results' subheading would help readers. Alternatively, this summary section could be moved to the Discussion and added to the first paragraph.

3) An additional Table/Figure describing the rich results could enhance the manuscript. Figure 1 gives the theoretical framework, but a table with the 3 themes and some bullets with big ideas would be a helpful addition.

6. PLOS authors have the option to publish the peer review history of their article (what does this mean?). If published, this will include your full peer review and any attached files.

Reviewer #1: **Yes: **Bethany Barone Gibbs

---

## [Author Response · Author response to Decision Letter 0]

21 Aug 2020

Revisions Requested by Editorial Office 08/19/2020

1. Thank you for including your ethics statement on the online submission form: 

"The study protocol was approved by the institutional review board at Brown University

(protocol #1412001171) on September 3, 2015. Participants provided written informed

consent.". 

To help ensure that the wording of your manuscript is suitable for publication, would you please also add this statement at the beginning of the Methods section of your manuscript file.

Response: Text has been added at the beginning of the METHODS section in the manuscript.

2. Thank you for providing a Data Availability Statement and explaining why the data are restricted. In order for your statement to meet our data sharing policy requirements (https://journals.plos.org/plosone/s/data-availability), however, we ask that you please provide a non-author, institutional point of contact (with contact information) that is able to field data access queries. PLOS ONE's data policy requires this in the interest of maintaining long-term data accessibility.

Response: Reseachdata@brown.edu will be added as an institutional point of contact. The data is made available upon submission of a data usage agreement to the Brown Digital Repository through this contact. 

3. Thank you for providing your revised Competing Interest Statement. We've made some updates to conform to journal policy, including adding some adherence statements. Can you please confirm whether the following proposed statement is accurate and suitable to appear alongside your manuscript?

"The authors have read the journal’s policy and have the following potential competing interests: EC is the Director of the Mindfulness Center at Brown University. The Mindfulness Center, a non-profit entity, has an Education Unit that provides mindfulness-based program delivery to the general public for fees. Dr. Loucks’s salary is not tied to quantity, or content, of programs offered through the Mindfulness Center, and this therefore does not alter our adherence to PLOS ONE policies on sharing data and materials. There are no patents, products in development or marketed products associated with this research to declare.”

If there are any errors or omissions in these statements please let us know. Otherwise, if the above statement is correct, please include it in your Cover Letter upon resubmission and we will update the statement in our system on your behalf, and proceed with the review process.

Response: The requested revision is acceptable to the authors and has been added to the revised cover letter. 

Revisions Requested by Reviewers 07/29/2020

Response: The competing interests statement has been added to the cover letter. 

Response: The following data sharing statement has been clarified and added to the cover letter.

The datasets presented in this article are not readily available because sensitive and potentially identifying information from this qualitative research, including original interviews, cannot be provided due to ethical restrictions. Requests to access the datasets should be directed to the principal investigator Dr. Eric Loucks (eric_loucks@brown.edu).

PLOS requires an ORCID iD for the corresponding author in Editorial Manager on papers submitted after December 6th, 2016. Please ensure that you have an ORCID iD and that it is validated in Editorial Manager. To do this, go to ‘Update my Information’ (in the upper left-hand corner of the main menu), and click on the Fetch/Validate link next to the ORCID field. This will take you to the ORCID site and allow you to create a new iD or authenticate a pre-existing iD in Editorial Manager. Please see the following video for instructions on linking an ORCID iD to your Editorial Manager account: https://www.youtube.com/watch?v=_xcclfuvtxQ

Reviewer #1: I enjoyed reading this manuscript that employed qualitative methods to gain in-depth understanding of how a mindfulness-based intervention could improve health behaviors related to elevated blood pressure. The manuscript is a joy to read, with clear and well-written text sections in a logical, easy-to-read flow. The context of the study in the phase-I trial is well explained. Overall, the study offers important information for understanding mechanisms by which the mindfulness intervention improves health behaviors that could lead to the substantial reduced BP observed in the phase I trial. The addition of the in-depth interviews to the focus groups is lovely (to reach further!). I have a few minor comments for improvement as follow:

1) Though the explanations are clear, more references for the qualitative methods used are needed. I note that 'The Qualitative Study' (lines 157-192) and the 'Data Analytic Approach' (lines 210-234) have limited or no references. Adding these references in would enhance the manuscript and provide readers with additional resources to understand the methods used (i.e., the qualitative research methodology).

Response: Thank you. Citations have been added to both sections consistent with the reviewers recommendations. 

Manuscript Change (pg. 6-9, line(s):156-237): 

The Qualitative Study 

Purposive sampling of intervention participants was used to recruit for the qualitative study.21 Intervention study participants were contacted by members of the research team to assess their willingness to participate. Participation was voluntary and independent of participation in the intervention trial. All study participants who completed the MB-BP program (n=48) were eligible to participate. 

First, three focus group discussions (FGDs) were conducted including 19 participants (FGD 1, n=4; FGD 2, n=8; FGD 3, n=7). The FGDs were conducted by a co-author (XX), an experienced qualitative researcher, assisted by a note taker (XX). Neither had a relationship with protocol development, implementation, intervention assessments, or contact with participants prior to the FGD. The FGDs were audio recorded and held in a private room at the Brown University School of Public Health in Providence, RI. FGDs occurred within two months after completion of the intervention. Each session lasted between 1.5-2 hours and was conducted in accordance with an approved focus group protocol see Supplementary Material File 1. 

FGD participants were provided with food and refreshments but were not compensated for participation. 

Seven semi-structured, in-depth interviews (IDIs) were conducted with participants who declined participating in the focus group discussions, in order to foster a more representative sample, and to more effectively capture individual perspectives and experiences with the intervention.19 They were selected using a randomized list of possible candidates by a senior member of the research staff. Those selected were then followed up to inquire about whether they would be willing to participate in an in-depth interview. Ten participants were contacted with three refusals in order to recruit the seven individuals included in the IDIs. All of the IDIs were conducted by phone, by a senior member of the research team. Participants were reimbursed a gift card ($25) for participating in the 25-minute IDI. 

In-Depth Interviews (IDI) 

The IDIs were conducted by a member of the study staff trained in qualitative methods under the guidance of the consulting qualitative expert (XX). The researcher conducting the interviews was familiar with MBPs and interested in conducting and evaluating MBPs. The interviews were conducted using Zoom (Version # 4.1.24407.0507, San Jose, CA.) software and audio recorded. The interviews were conducted in accordance with an approved protocol see Supplementary Material File 2. 

Intervention Description

This study adapted Mindfulness-Based Stress Reduction (MBSR) for participants with prehypertension/hypertension, creating MB-BP. Specifically, MB-BP is based on, and time-matched to, the standardized MBSR intervention which consists of eight 2.5-hour weekly group sessions, and a 7.5-hour one-day sessions, and is described elsewhere.5,22-25 MB-BP and MBSR contain similar instruction and practices in mindfulness meditation and conversations about stress and coping. 

The unique areas of MB-BP are education on hypertension risk factors, hypertension health effects, and specific mindfulness modules focused on awareness of diet, physical activity, medication adherence, alcohol consumption, stress, and social support for behavior change. MB-BP builds a foundation of mindfulness skills (e.g. meditation, yoga, self-awareness, attention control, emotion regulation; Figure 1) through the modified MBSR curriculum. MB-BP then directs those skills towards participants’ relationship with their risk factors for hypertension. Details concerning study design, intervention assessments, and preliminary acceptability and feasibility results as well as customizations of MB-BP from MBSR are found in further detail elsewhere.17 The curriculum guide and MB-BP instructor certification program can be accessed by contacting the lead author. 

Data Analytical Approach 

Trained research assistants reviewed audio recordings and transcribed recordings verbatim. The data was collected as part of a larger mixed methods study and was analyzed using thematic analysis.5,26-28 Thematic analysis was considered the most appropriate analytical approach for the semi-structured data from the focus group discussion and in-depth interviews. 

Both FGD and IDI transcripts were double-coded by two members of the research team who then identified both a priori and emergent codes by two members of the research team using NVivo v.11.27,28 Cross checks for coding consistency were performed by a third member of the research team (XX). Codes with definitions are listed in the qualitative codebook (see Supplementary Material 3). The data are available from the corresponding author upon request with ethics approval. 

The team met under the supervision of the qualitative expert (XX) during a series of three in-person meetings to: (1) cluster FGD coding into broad themes, (2) discuss themes and additional qualitative work needed, and (3) refine FGD and IDI themes into main themes and sub-themes. During the iterative process of data analysis for the FGDs, the team found that saturation had not been achieved. In-depth interviews were planned to gain further insights into the main research questions, as well as to gain an individual perspective to enhance the group-based collective understanding of the FGDs. Videoconference- and phone-based IDI’s provide an additional benefit to reduce possible biases associated with focus group participation, where it is plausible that those who had less positive experiences in the MB-BP course may have declined coming to the in-person focus groups.29 In addition to the importance of the individual perspective gained from the IDIs, participant burden for phone/videoconference IDI’s is less and participation rates are higher, and so it was thought that this could provide a more representative sample overall.29,30 The coding structure from the FGD was used to inform the coding of the IDI's. Discrepancies were resolved through group consensus and communications between the coders (XX, XX, XX) and the qualitative expert (XX) with input from the content expert when applicable (XX). The results of this process and the integrated thematic analysis of qualitative data from both FGDs and IDIs are presented below. 

2) The results are lovely, but would benefit from an additional subtitle to separate the overall summary at the end. As it is, the 3rd theme just bleeds into the overall - an 'Overall findings' or 'Summary of Results' subheading would help readers. Alternatively, this summary section could be moved to the Discussion and added to the first paragraph.

Response: Final section has been separated with heading “Overall Summary” consistent with the reviewers recommendation. 

Manuscript Change (pg. 19, line(s). 422-435): 

Overall Summary

For many participants the program led to important changes in their health behaviors after becoming more self-aware and for some these changes were made in multiple areas of their lives. This is best captured by one participant who described how she was able to exert greater control over her life, stress responses, eating behaviors, and anxiety to lower her BP: “Being mindful of what makes us anxious and trying to control our reaction to what makes us anxious and, in addition, we were also taught to be mindful of what makes us happy. So, to reinforce different things that make you happy, and also with the eating, reinforce thinking about how your body feels when you eat the foods that maybe aren’t making your body feel good versus the foods that are more helpful. So, I think it affected everything in my whole life.”(F2). 

In summary, participants indicated that the program impacted their cardiovascular health directly through becoming more aware of their present moment experiences, allowing them to make positive choices for their health behaviors, specifically minimizing stress reactivity and making healthier choices concerning their diet. 

3) An additional Table/Figure describing the rich results could enhance the manuscript. Figure 1 gives the theoretical framework, but a table with the 3 themes and some bullets with big ideas would be a helpful addition.

Response: We appreciate this recommendation. The following table has been created and added to the manuscript:

Manuscript Change (pg. 12): 

Table 2. Qualitative themes outlining preliminary findings mechanisms by which MB-BP may influence blood pressure. Examples of participants’ quotes are shown. 

Academic Editor: This is a methodologically strong and well-written manuscript reporting interesting new qualitative findings on the possible processes underlying positive effects of mindfulness on cardiovascular health. Minor amendments to the manuscript are needed before it can be accepted for publication in PLOS ONE. I agree with the comments from Reviewer 1, please make sure to address these in your revision. I have a couple of additional comments that need addressing in the revision:

1.'craving' is included as a possible mechanisms of mindfulness on line 79, this should be corrected or clarified

Response: Craving has been removed from the sentence. 

Manuscript Change (pg. 3, line(s): 78-80): 

“Major proposed mechanisms include decentering, self-compassion, acceptance, self-awareness, attention control, and emotion regulation.”

2.I don't think 'self-awareness' and 'interoception' are equivalent terms, 'interoception' describes sensation-related awareness whereas 'self-awareness' is a broader term involving awareness of self-construal related processes (lines 89-90), please clarify why you think the two terms can be used interchangeably or don't use them in this way

Response: “or interoception” has been removed. 

Manuscript Change (pg. 4, line(s). 89-90): 

“Major proposed mechanisms include decentering, self-compassion, acceptance, self-awareness, attention control, and emotion regulation.9-13 Rigorous testing of the mechanisms using mediation analyses is lacking.12,13”

3.Please explain briefly why the thematic analysis approach has been chosen and support the explanation by relevant references

Response: Manuscript has been changed see below. 

Manuscript Change (pg. 9, line(s). 211-213): 

The data was collected as part of a larger mixed methods study and was analyzed using thematic analysis.5,26-28 Thematic analysis was considered the most appropriate analytical approach for the semi-structured data from the focus group discussion and in-depth interviews.

---

## [Editor Report · Decision Letter 1]

9 Sep 2020

Mindfulness and Cardiovascular Health: Qualitative Findings on Mechanisms from the Mindfulness-Based Blood Pressure Reduction (MB-BP) Study

PONE-D-20-16708R1

Dear Dr. Nardi,

We’re pleased to inform you that your manuscript has been judged scientifically suitable for publication and will be formally accepted for publication once it meets all outstanding technical requirements.

Kind regards,

Dusana Dorjee, PhD

Academic Editor

PLOS ONE

Additional Editor Comments (optional):

Please remove research participant age information from Table 2 since this is a partial identifier, this is what is currently listed in the table for one of the participants: '(female, 58 years old, FGD)'.
---

## [Editor Report · Acceptance letter]

11 Sep 2020

PONE-D-20-16708R1 

Mindfulness and Cardiovascular Health: Qualitative Findings on Mechanisms from the Mindfulness-Based Blood Pressure Reduction (MB-BP) Study 

Dear Dr. Nardi:

I'm pleased to inform you that your manuscript has been deemed suitable for publication in PLOS ONE. Congratulations! Your manuscript is now with our production department. 

Kind regards, 

on behalf of

Dr. Dusana Dorjee 

Academic Editor

PLOS ONE